# Limited Knowledge of Toxoplasmosis-Specific Preventive Behaviors in Pregnant Women: A Survey Study in Northern Italy

**DOI:** 10.3390/ijerph22040517

**Published:** 2025-03-28

**Authors:** Federica Fernicola, Elisabetta Colciago, Giulia Capitoli, Elisa Merelli, Francesca Arienti, Valeria Poletti De Chaurand, Gaia Scandella, Anna Carli, Sara Ornaghi

**Affiliations:** 1Department of Obstetrics, Fondazione IRCCS San Gerardo dei Tintori, 20900 Monza, Italy; arienti.francesca@outlook.com (F.A.); v.polettidechaura@campus.unimib.it (V.P.D.C.); g.scandella5@campus.unimib.it (G.S.); annacarli88@gmail.com (A.C.); sara.ornaghi@unimib.it (S.O.); 2Department of Medicine and Surgery, University of Milano-Bicocca, 20900 Monza, Italy; elisabetta.colciago@unimib.it; 3Bicocca Bioinformatics Biostatistics and Bioimaging B4 Center, School of Medicine and Surgery, University of Milano-Bicocca, 20854 Vedano al Lambro, Italy; giulia.capitoli@unimib.it (G.C.); e.merelli1@campus.unimib.it (E.M.); 4Biostatistics and Clinical Epidemiology, Fondazione IRCCS San Gerardo dei Tintori, 20900 Monza, Italy

**Keywords:** toxoplasmosis, infection, congenital, knowledge, pregnancy

## Abstract

Toxoplasmosis (TG) is a zoonotic disease that, if contracted during pregnancy, can lead to fetal infection with severe complications. Currently, the only way to prevent maternal infection during pregnancy is to adopt hygiene practices. Our study aimed to evaluate pregnant women’s awareness of TG infection, their knowledge of TG-specific preventive hygiene measures, and factors linked to inadequate knowledge as potential targets for intervention. A prospective survey study was conducted at a university hospital between May and November 2023. A self-administered questionnaire was given to pregnant women. Out of 402 participants, 95.3% were aware that TG could be a potential issue in pregnancy. However, only 22.5% of them were able to identify all four correct hygiene behaviors to prevent TG infection acquisition during gestation. Women with a higher level of education, of European origin, and who had heard of other potentially serious conditions in pregnancy were more likely to have an appropriate level of knowledge about TG. Healthcare professionals and mass media, when employed as sources of information before conception, played a positive role in enhancing pregnant women’s awareness of TG. In conclusion, while pregnant women showed high levels of awareness of TG infection, their knowledge of preventive measures was limited, with only less than one-fourth with appropriate knowledge. This highlights the need for innovative strategies, possibly face-to-face interactions, digital tools, and public health campaigns, to improve the ability of healthcare professionals to deliver accurate and accessible information to reduce the risk of infection acquisition during gestation.

## 1. Introduction

Toxoplasmosis, caused by the protozoan parasite Toxoplasma gondii (TG), is a zoonotic disease affecting approximately one-third of the global population [1].

In immunocompetent individuals, TG infection is often asymptomatic or causes mild flu-like symptoms. However, if contracted during pregnancy, it can be transmitted transplacentally, leading to congenital infection. Transmission during the first trimester may result in adverse outcomes, such as miscarriage or stillbirth, whereas transmission in the second trimester can lead to neonatal birth defects, including neurocognitive impairment and ocular disease [2].

The global incidence of congenital TG is 0.15 cases per 10,000 live births [3]. Within Europe, the EUROTOXO program [4] reports varying incidences across nations, ranging from 0.7 to 13 cases per 10,000 births in Sweden and Germany, respectively.

Due to the absence of a vaccine, TG infection acquisition in seronegative pregnant women can only be prevented through hygiene practices and healthy behaviors [5]. Preventive measures include dietary recommendations, such as consuming only thoroughly cooked meat, washing raw vegetables, avoiding raw eggs or unpasteurized milk, wearing gloves while changing cat litter or during gardening, and thoroughly cleaning hands and nails afterward [6,7,8].

Extensive dissemination of information about TG among pregnant women has been proven to significantly reduce seroconversion rates and encourage behavioral changes [9,10]. Notably, Italian guidelines for low-risk pregnancies recommend providing information on preventing TG infection to all pregnant women, those planning to conceive, and their partners [7].

Awareness of TG infection during pregnancy has been evaluated in various studies, with different findings across countries [11,12,13,14]. In the United States, Jones et al. found that only 48% of the pregnant women surveyed had received information about toxoplasmosis [12]. In Europe, awareness levels were reported to be higher, with 75% to 95% of pregnant women being familiar with the potential risks related to TG infection [11,13,15].

One Italian cross-sectional study in non-pregnant women found that 84% of them were aware of TG infection [16]. However, the understanding of the condition was frequently superficial, with only few women being able to accurately identify the necessary measures for preventing TG infection acquisition [16]. Additionally, this study failed to consider the potential factors contributing to this lack of knowledge.

Our study aimed to assess pregnant women’s awareness of TG infection and knowledge of specific hygiene measures to prevent infection acquisition; also, we aimed to accurately investigate factors associated with inadequate knowledge, which represents a potential target of intervention.

## 2. Materials and Methods

A prospective cross-sectional study was conducted from 15 May to 15 November 2023, at a University Maternity Department in Northern Italy, which handles approximately 2600births annually. The hospital is located in an industrial area, and 20% of the patients who access its services are of non-Caucasian origin. Data collection utilized a self-administered paper questionnaire (Appendix A), developed following a review of the relevant literature [11,13,15] and consultation with specialists involved in the care of pregnant women affected by TG infection.

Before distributing the survey to the entire sample, a pilot test was conducted with a small group of 20 pregnant women, selected to ensure heterogeneity in age, pregnancy trimester, nationality, and educational background. Experts in fetal medicine were consulted to evaluate the clarity, relevance, and comprehensiveness of the questionnaire, identifying any ambiguous or misleading questions. To assess internal consistency, a test–retest reliability analysis was performed.

Participants included pregnant women aged 18 years or older and receiving antenatal care at the Maternity Unit. Anonymity was maintained to encourage candid responses. The questionnaire took approximately 10 min to complete.

The questionnaire, distributed to women in any trimester, consisted of multiple-choice questions divided into two sections. The first section covered sociodemographic details (such as age, education, employment, nationality, number of previous children, and expected birth date). The second section focused on behaviors during pregnancy, including alcohol consumption, smoking habits, folic acid supplementation, awareness of mandatory vaccinations during pregnancy, knowledge of TG infection and its potential consequences in pregnancy, familiarity with TG-specific preventive measures, and sources of information regarding TG infection and its prevention.

According to the Italian guideline for low-risk pregnancies, universal screening for TG is recommended for all pregnant women in the first trimester of gestation, and every 4–6 weeks thereafter in seronegative women [7].

At our institution, midwives educate pregnant women during their first antenatal appointment in the first trimester regarding perinatal infections, including TG, and the related preventive behaviors.

### Statistics

The minimum required sample size was 384 participants, calculated using the Creative Research System’s Sample Size Calculator [17]. The target population size was based on the number of live births in 2023 as reported by the Italian National Institute of Statistics [18]. The expected frequency (outcome probability) was assumed to be 48% based on previous studies that reported awareness about TG infection among pregnant women ranging between 48% and 95% [11,12,13,15].

Knowledge of TG preventive measures was categorized into three levels: (1) “appropriate” if women provided all four correct answers and no incorrect responses; (2) “moderate” if women gave three correct answers without any wrong ones or four correct answers with one incorrect response; (3) “minimal” if any other combination of correct and incorrect responses was given, not fitting into the previous two categories.

A chi-square test and ANOVA were used to make comparisons between these three knowledge categories in terms of qualitative and continuous variables, respectively. Univariate logistic models were used to assess the impact of different factors on the awareness of congenital TG infection in pregnancy and knowledge of its preventive measures.

A *p*-value of less than 0.05 was considered statistically significant, indicating that at least one group’s median knowledge level differed from that of the other groups.

Statistical analyses were performed using the open-source R software v.4.4.2 (R core Team. A language and environment for statistical computing [Software]. R Foundation for Statistical Computing).

This study was approved by the Brianza Ethics Committee (N. 4191, 9 March 2023). Following a thorough explanation of the study by the researchers, written informed consent was obtained from the participants, who could decline or withdraw at any time.

## 3. Results

A total of 426 questionnaires were collected. Of those, 24 were excluded because they were only partially completed by the participants, thus leading to a final study population of 402 women.

Figure 1 shows women’s general awareness of various conditions relevant to gestation. Overall, 377 (93.8%) women were aware of the existence of toxoplasmosis, making it the most commonly recognized condition.

TG infection was recognized as a potential issue during pregnancy by 383 (95.3%) women. However, only 86 (22.5%) of them displayed a comprehensive understanding of the infection, correctly identifying all four preventive behaviors listed in the questionnaire (Figure 2). A moderate level of knowledge was identified in 134 (35.0%) women, whereas 163 (42.5%) showed minimal knowledge.

Among the 383 participants who recognized TG as a potential problem in pregnancy, washing uncooked fruits and vegetables before consumption was the most commonly identified preventive measure against TG infection acquisition (*n* = 357, 93.2%), followed by avoiding undercooked meat (*n* = 347, 90.6%). Conversely, taking precautions while handling material potentially contaminated by cat feces was the least recognized preventive behavior (*n* = 228, 59.5%) (Figure 2).

Participants’ responses to the questionnaire categorized by the level of knowledge regarding TG-specific preventive behaviors are reported in Table 1.

More than half of the participants were in their first pregnancy (*n* = 205, 53.5%) and had a university education (*n* = 217, 57.7%).

A total of 94.1% (*n* = 354) of the women were of European origin and completed the questionnaire while in their third trimester (*n* = 202, 60.5%).

The mean maternal age was 33.34 (±4.48 standard deviation, SD). Almost all participants (*n* = 372, 97.1%) reported receiving information on healthy lifestyles and preventive measures to avoid perinatal infection acquisition, primarily during their first antenatal appointment (*n* = 280, 75.9%).

Women with a higher level of education (*p* = 0.025), of European origin (*p* = 0.040), and who had heard of other potentially serious conditions in pregnancy, such as cytomegalovirus (*p* = 0.008), spina bifida (*p* = 0.008), parvovirus B19 (*p* = 0.033), fetal alcohol spectrum disorders (*p* = 0.001), Down syndrome (*p* = 0.027), HIV (*p* = 0.010), and syphilis (*p* = 0.040), were more likely to have an appropriate level of knowledge about TG (Table 2). Additionally, awareness that parvovirus (*p* = 0.008) and cytomegalovirus (*p* = 0.001) could pose specific risks during pregnancy was also associated with appropriate knowledge of TG.

Among the 383 women who recognized TG as a potential concern during pregnancy, 312 (81.5%) reported having received information about preventive measures before conception. Among these 312 women, healthcare providers were identified as the primary source of this information (*n* = 140, 44.9%), followed by family members (*n* = 101, 32.4%) and mass media (*n* = 99, 31.7%), with friends being the least cited source (*n* = 47, 15.1%). Of note, healthcare providers were the primary source of information also among women with minimal knowledge of TG-specific preventive behavior.

The spaghetti plot (Figure 3) illustrates the relationship between sources of information used by participants before conception and women’s age. Up to the age of 30–34 years, healthcare providers and family members were the primary sources. However, after this age range, the influence of family members declined gradually, whereas healthcare providers remained predominant. Interestingly, mass-media usage increased until age 35 and then stabilized.

Figure 4A,B display the relationship between sources of information, both individually and in various combinations, adopted by TG-aware women before conception and their level of knowledge about specific hygiene measures. Women who primarily received information from healthcare providers and mass media demonstrated a higher level of knowledge (depicted in green); conversely, reliance on family and friends was associated with minimal knowledge (depicted in red) (Figure 4A).

## 4. Discussion

Our cross-sectional single-center study aimed to assess awareness of TG infection and knowledge of its specific preventive measures and to identify relevant associated factors among pregnant women managed at a Northern Italian academic maternity center. The findings reveal a substantial gap between awareness and knowledge, highlighting the importance of targeted educational interventions to mitigate the risks associated with TG infection acquisition during pregnancy.

Our data show that almost all participants (95.3%) recognized TG infection as a potential issue during pregnancy. This figure is higher than those previously reported in other studies on TG infection awareness in European countries, such as Switzerland (87%) and the Netherlands (75.3%) [13,15], and similar to more recently published data from Poland [11].

A potential explanation could be that our participants had received specific counseling regarding hygiene measures in pregnancy during the first trimester, along with detailed leaflets on the topic. In addition, the presence of a decade-long history in our country regarding prevention programs for congenital toxoplasmosis may have played a role in our findings [19]. Starting in 1999, the National Institute of Health has emphasized the importance of primary prevention through educational initiatives targeting both healthcare professionals and pregnant women [20]. Also, our data may have been affected by the execution of monthly serological screening for TG infection in seronegative pregnant women, which has been recommended and performed on a national level since 2010 [21]. Furthermore, almost 60% of the respondents had a university degree, thus suggesting that a high level of education might be associated with heightened awareness levels, as previously reported [12,13,14,15,22]. Of note, no prior Italian studies explored pregnant women’s awareness of TG infection. The only available Italian study assessed non-pregnant women and only evaluated “having heard about Toxoplasmosis”, without providing any data regarding awareness of potential consequences of TG infection acquisition during gestation and knowledge of its preventive hygiene behaviors [16].

Despite the high level of awareness identified in our study population, only 22.5% of women aware of TG could correctly identify all four preventive behaviors, thus indicating that awareness does not necessarily translate into comprehensive knowledge. This aligns with previous studies that reported a superficial understanding of TG infection among pregnant women despite high levels of awareness [11,12,13,15].

Almost half of the respondents identified only one or two correct behaviors to avoid TG infection, as shown in Figure 2. The most frequently recognized preventive measures were washing uncooked fruits and vegetables before consumption and avoiding undercooked meat. These results are consistent with a previous publication on the topic [23] and the fact that one of the main sources of TG infection in Italy is the consumption of undercooked meat [24]. In turn, taking precautions while handling material potentially contaminated by cat feces was the least recognized preventive behavior. Notably, a recent Italian study has reported a significant rise in TG infection rates in pregnancy during the 2020 SARS-CoV2 pandemic, likely associated with increased cat ownership [25]. Our data revealed that respondents with higher education levels were significantly more likely to have appropriate knowledge of toxoplasmosis, which aligns with existing studies indicating the importance of education in health literacy [11,12,13,14,22,26].

Furthermore, women familiar with other infections posing a risk during gestation, such as cytomegalovirus or parvovirus B19, were more likely to understand toxoplasmosis, emphasizing the role of comprehensive education in improving appropriate implementation of preventive measures. Evidence suggests that the level of overall knowledge of women about antenatal care has a significant positive correlation with their practices during pregnancy [27].

Interestingly, almost all women (81.5%) reported having received information before pregnancy about hygiene measures and appropriate behaviors to follow, and in half of the cases, the first source of information was healthcare professionals.

Previous studies have reported that 85–100% of healthcare professionals counsel pregnant women about toxoplasmosis [12,13]. However, the effectiveness of this education depends on several factors, including the clarity of communication, the frequency of educational sessions, and the use of education strategies [28]. Repeated and reinforced education from healthcare providers can significantly improve pregnant women’s understanding of and adherence to preventive measures [13]. Also, involving partners in educational sessions can empower parents and help reinforce the practices at home [29]. A study by Karamolahi et al. [30] found that interactive educational tools, such as mobile apps, substantially increase the health literacy of pregnant women, thus suggesting that mobile-based interventions can be an option for mass dissemination. Our findings confirm that digital tools play a complementary role alongside healthcare professionals in effectively increasing women’s knowledge levels. Of note, it is essential that these mobile apps are either designed or, at a minimum, verified by healthcare professionals. While TG infection during the first and second trimesters is linked to unfavorable pregnancy outcomes, it remains crucial to educate women about preventive measures before conception and during pregnancy. Public health campaigns play a strategic role in promoting and expanding knowledge while encouraging the target population to adopt beneficial attitudes and behaviors [13]. Notably, in recent years, the role of apps in improving communication has been emerging [31]. Digital health tools, such as mobile apps, online platforms, and social media, offer innovative ways to provide pregnant women with accessible, interactive information and real-time support [32]. These tools can enhance knowledge, motivate behavior change, and foster community engagement while enabling healthcare professionals to address misconceptions and share accurate information [33]. With widespread smartphone and internet use, especially among younger populations, these strategies can be highly effective for promoting TG infection prevention [30]. On the other hand, it is important to recognize that while medical apps are not entirely unregulated, app-specific legislation remains scarce [34]. Moreover, despite the wide availability of pregnancy-related apps, only a few are of high quality and provide evidence-based content [35,36].

This study presents several strengths, including being the first of its kind in Italy to assess knowledge about toxoplasmosis prevention among pregnant women. Additionally, the sample size is adequate, providing a solid foundation for drawing meaningful conclusions. Our study focused on pregnant women in Northern Italy, but its findings may be relevant to other countries where TG infection represents a substantial public health concern, particularly in geographical areas with universal screening programs and high TG prevalence, such as France, Spain, Belgium, and Germany.

However, some limitations should be acknowledged. First, this study was conducted at a single research site in a non-rural area, which may limit the generalizability of the findings to other regions or healthcare settings. Furthermore, 94% of the participants were of European origin, potentially introducing bias and reducing the applicability of the results to more diverse populations. Another limitation is the potential influence of specific counseling provided early in pregnancy, which could have increased the level of awareness among participants, particularly those who received targeted information during their first antenatal appointment. This counseling may have positively impacted their knowledge, leading to results that might not reflect the broader population of pregnant women who do not receive the same level of care. In addition, self-reported data collected through questionnaires are susceptible to recall bias, particularly regarding past information exposure, potentially affecting the accuracy of responses. Also, multiple-choice questions may not fully capture the complexity of health behavior change, which could be better explored through open-ended questions. Finally, the study design did not include a control group (e.g., non-pregnant women of childbearing age) to establish a baseline for TG awareness and knowledge within the general population. Future studies could address this limitation by incorporating a control group for comparative analysis.

## 5. Conclusions

In conclusion, while pregnant women demonstrated a high level of awareness about TG, their understanding of specific preventive measures and behaviors was superficial, with nearly half of the study population displaying a minimal level of knowledge. We advocate for the implementation of innovative communication strategies that integrate traditional face-to-face interactions with digital tools and public health campaigns. By integrating technology and diverse communication strategies, healthcare providers can ensure that pregnant women receive comprehensive, accurate, and easily accessible information about TG infection prevention, ultimately reducing the risk of infection and its associated complications.

Future studies could explore the effectiveness of different educational interventions, such as mobile health applications, targeted social media campaigns, and structured prenatal education programs, in enhancing TG awareness and preventive behaviors. Additionally, longitudinal studies assessing the impact of improved awareness on actual behavioral changes and infection rates would provide valuable insights.

## Figures and Tables

**Figure 1 ijerph-22-00517-f001:**
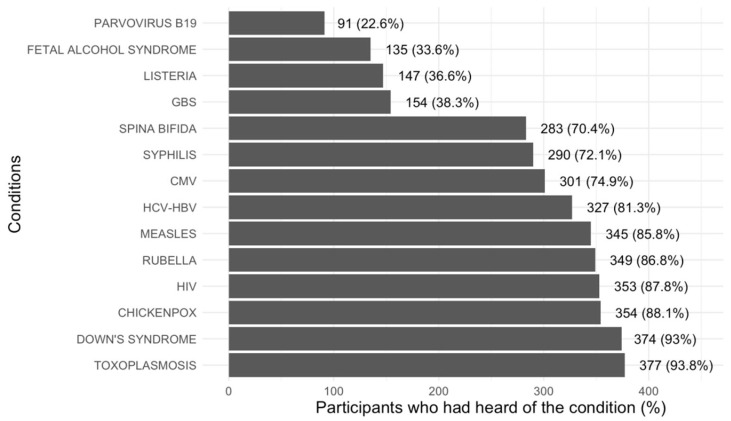
Awareness of conditions investigated by the questionnaire. GBS: Group B Streptococcus; CMV: cytomegalovirus; HCV: hepatitis C virus; HBV: hepatitis B virus; HIV: human immunodeficiency virus.

**Figure 2 ijerph-22-00517-f002:**
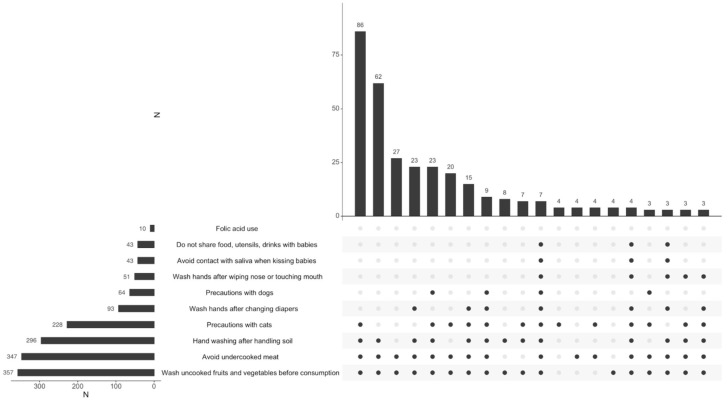
Women’s knowledge of specific hygiene measures to avoid acquisition of TG infection in pregnancy; N: number. The questionnaire allowed women to give one or more answers. The last four listed behaviors are those specific for preventing TG infection acquisition. The vertical bars identify the number of women who chose the combination of answers marked with the black dots underneath. The figure shows the combination of answers chosen by at least three women. The horizontal bars identify the total number of women who gave the specific answer reported.

**Figure 3 ijerph-22-00517-f003:**
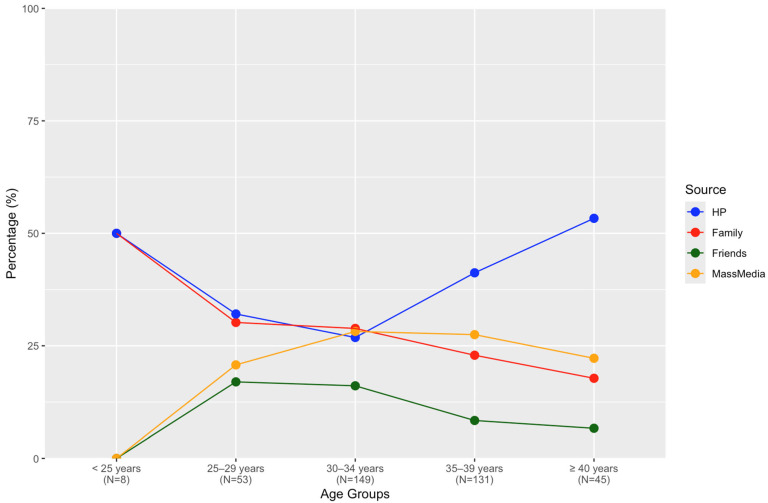
Sources of information used before conception according to respondents’ ages. HP: healthcare provider.

**Figure 4 ijerph-22-00517-f004:**
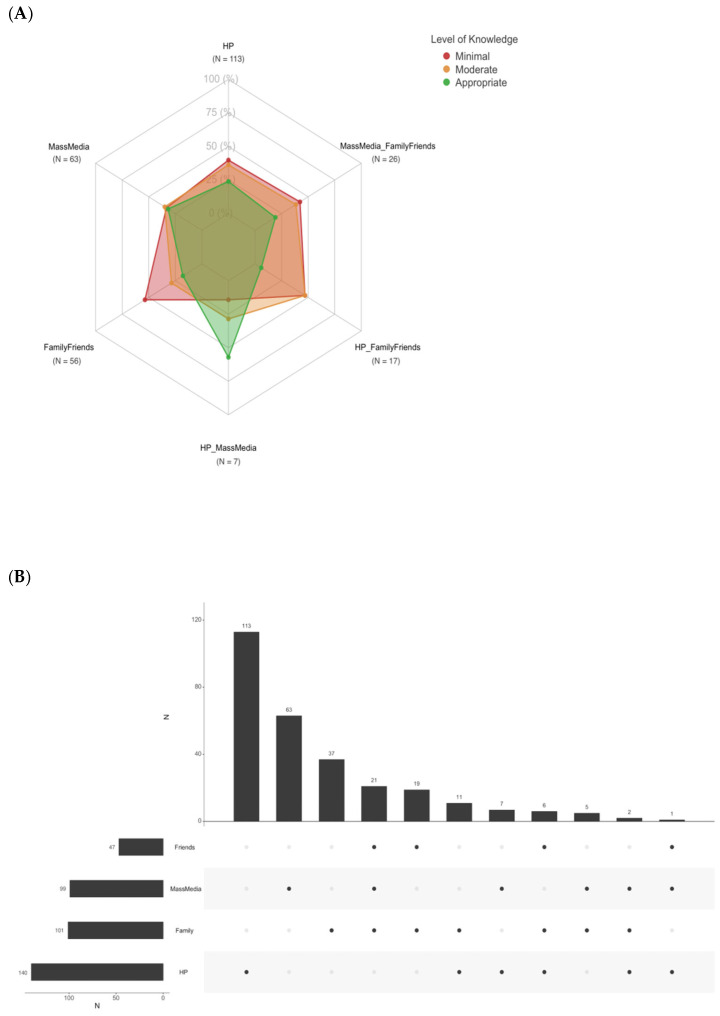
(**A**,**B**) Knowledge of TG-specific preventive measures based on sources of information, both individually and in various combinations, received before conception by TG-aware participants (*n* = 312). The Family Friends category includes women who reported obtaining information from family members or friends or both (**A**). Women’s specific sources of information before conception; N: number. Women were allowed to give one or more answers. The vertical bars identify the number of women who chose the combination of answers marked with the black dots underneath. The figure shows the combination of answers chosen by at least one woman. The horizontal bars identify the total number of women who gave the specific answer reported (**B**). HP: healthcare provider.

**Table 1 ijerph-22-00517-t001:** Participants’ answers to the questionnaire according to knowledge levels. General characteristics and data on information regarding preventive measures.

		Overall (*n* = 383)	Minimum Level of Knowledge (*n* = 163; 42.5%)	Moderate Level of Knowledge (*n* = 134; 35.0%)	Appropriate Level of Knowledge (*n* = 86; 22.5%)	Overall *p*-Value *	1. *p*-Value Minimun vs. Moderate 2. *p*-Value Minimun vs. Appropriate 3. *p*-Value Moderate vs. Appropriate
Education level	Primary school	21 (5.6)	9 (5.7)	11 (8.4)	1 (1.2)	0.025	1. 0.932 2. 0.005 3. 0.005
	Secondary school	138 (36.7)	66 (41.5)	48 (36.6)	24 (27.9)		
	University	217 (57.7)	84 (52.8)	72 (55.0)	61 (70.9)		
At least 1 previous child	No	205 (53.5)	89 (54.6)	75 (56.0)	41 (47.7)	0.454	
	Yes	178 (46.5)	74 (45.4)	59 (44.0)	45 (52.3)		
Maternal age (mean ± SD)		33.34 ± 4.48	33.31 ± 4.40	33.54 ± 4.78	33.09 ± 4.17	0.766	
Maternal age	<25 years	10 (2.5)	4 (2.2)	2 (1.5)	4 (4.7)	0.126	
	25–29 years	52 (13.0)	25 (14.0)	22 (16.2)	5 (5.8)		
	30–34 years	156 (38.9)	70 (39.1)	45 (33.1)	41 (47.7)		
	35–39 years	133 (33.2)	59 (33.0)	45 (33.1)	29 (33.7)		
	≥40 years	50 (12.5)	21 (11.7)	22 (16.2)	7 (8.1)		
Nationality	European	354 (94.1)	144 (90.6)	127 (96.9)	83 (96.5)	0.040	1. 0.021 2. 0.058 3. 0.893
	Non-European	22 (5.9)	15 (9.4)	4 (3.1)	3 (3.5)		
Data collection time	1st trimester	41 (12.3)	19 (14.0)	15 (12.9)	7 (8.5)	0.765	
	2nd trimester	91 (27.2)	34 (25.0)	33 (28.4)	24 (29.3)		
	3rd trimester	202 (60.5)	83 (61.0)	68 (58.6)	51 (62.2)		
Antenatal care setting	Academic maternity center	289 (75.9)	131 (80.9)	94 (70.7)	64 (74.4)	0.103	
	Community	24 (6.3)	5 (3.1)	13 (9.8)	6 (7.0)		
	First-level hospital	9 (2.4)	2 (1.2)	6 (4.5)	1 (1.2)		
	Private HP	59 (15.5)	24 (14.8)	20 (15.0)	15 (17.4)		
Information on preventive measures before conception	No	71 (18.5)	34 (20.9)	25 (18.7)	12 (14.0)	0.411	
	Yes	312 (81.5)	129 (79.1)	109 (81.3)	74 (86.0)		
Information on preventive measures during pregnancy	No	11 (2.9)	6 (3.7)	3 (2.2)	2 (2.3)	0.716	
	Yes	372 (97.1)	157 (96.3)	131 (97.8)	84 (97.7)		
When the information on preventive measures has been received	1st antenatal appointment	280 (75.9)	114 (73.1)	104 (80.0)	62 (74.7)	0.605	
	1st scan	54 (14.6)	26 (16.7)	14 (10.8)	14 (16.9)		
	At one antenatal appointment during pregnancy	35 (9.5)	16 (10.3)	12 (9.2)	7 (8.4)		
	Yes	288 (75.6)	114 (70.4)	101 (75.9)	73 (84.9)		

Data expressed as mean ± standard deviation (SD) and number over total (n/ntot) (%). Significance: χ^2^ analysis and Student’s *T* test; * *p* < 0.05. HP: healthcare professional.

**Table 2 ijerph-22-00517-t002:** Participants’ answers to the questionnaire according to knowledge levels. Questions regarding healthy habits and awareness of various conditions relevant to gestation.

		Overall (*n* = 383)	Minimum Level of Knowledge (*n* = 163; 42.5%)	Moderate Level of Knowledge (*n* = 134; 35.0%)	Appropriate Level of Knowledge (*n* = 86; 22.5%)	Overall *p*-Value *	1. *p*-Value Minimun vs. Moderate 2. *p*-Value Minimun vs. Appropriate 3. *p*-Value Moderate vs. Appropriate
Use of folic acid	No	58 (15.1)	25 (15.3)	15 (11.2)	18 (20.9)	0.144	
	Yes	325 (84.9)	138 (84.7)	119 (88.8)	68 (79.1)		
Use of folic acid	Before pregnancy	171 (52.6)	73 (52.9)	60 (50.4)	38 (55.9)	0.292	
	Following positive pregnancy test	104 (32.0)	43 (31.2)	36 (30.3)	25 (36.8)		
	Following 1st antenatal appointment	50 (15.4)	22 (15.9)	23 (19.3)	5 (7.4)		
Smoking in pregnancy	No	349 (91.8)	152 (93.3)	118 (89.4)	79 (92.9)	0.444	
	Yes	31 (8.2)	11 (6.7)	14 (10.6)	6 (7.1)		
Alcohol in pregnancy	No	341 (89.0)	147 (90.2)	123 (91.8)	71 (82.6)	0.084	
	Yes	42 (11.0)	16 (9.8)	11 (8.2)	15 (17.4)		
Heard about toxoplasmosis	No	11 (2.9)	7 (4.3)	3 (2.3)	1 (1.2)	0.318	
	Yes	370 (97.1)	155 (95.7)	130 (97.7)	85 (98.8)		
Heard about CMV	No	84 (22.0)	48 (29.6)	23 (17.3)	13 (15.1)	0.008	1. 0.011 2. 0.009 3. 0.703
	Yes	297 (78.0)	114 (70.4)	110 (82.7)	73 (84.9)		
Heard about spina bifida	No	100 (26.2)	52 (32.1)	36 (27.1)	12 (14.0)	0.008	1. 0.325 2. 0.002 3. 0.030
	Yes	281 (73.8)	110 (67.9)	97 (72.9)	74 (86.0)		
Heard about parvovirus B19	No	291 (76.4)	133 (82.1)	100 (75.2)	58 (67.4)	0.033	1. 0.163 2. 0.010 3. 0.186
	Yes	90 (23.6)	29 (17.9)	33 (24.8)	28 (32.6)		
Heard about fetal alcohol spectrum disorders	No	247 (64.8)	119 (73.5)	86 (64.7)	42 (48.8)	0.001	1. 0.111 2. <0.001 3. 0.015
	Yes	134 (35.2)	43 (26.5)	47 (35.3)	44 (51.2)		
Heard about Down syndrome	No	18 (4.7)	13 (8.0)	4 (3.0)	1 (1.2)	0.027	1. 0.043 2. 0.015 3. 0.528
	Yes	363 (95.3)	149 (92.0)	129 (97.0)	85 (98.8)		
Heard about HIV	No	34 (8.9)	22 (13.6)	10 (7.5)	2 (2.3)	0.010	1. 0.068 2. 0.003 3. 0.185
	Yes	347 (91.1)	140 (86.4)	123 (92.5)	84 (97.7)		
Heard about syphilis	No	93 (24.4)	48 (29.6)	32 (24.1)	13 (15.1)	0.040	1. 0.266 2. 0.011 3. 0.131
	Yes	288 (75.6)	114 (70.4)	101 (75.9)	73 (84.9)		

Data expressed as mean ± standard deviation (SD) and number over total (n/ntot) (%). Significance: χ^2^ analysis and Student’s *T* test; * *p* < 0.05.; CMV: cytomegalovirus; HIV: human immunodeficiency virus.

## Data Availability

Data are available upon reasonable request to the corresponding author.

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
