# Peer review of "Limited Knowledge of Toxoplasmosis-Specific Preventive Behaviors in Pregnant Women: A Survey Study in Northern Italy"

_ijerph, 2025, doi:10.3390/ijerph22040517_

Round 1

Reviewer 1 Report

Comments and Suggestions for Authors

Toxoplasmosis is parasitic infection associated with a wide range of possible congenital malformations if acquired during pregnancy. Prevention measures are focused on serological screening of women during pregnancy or before conception and on promotion of Toxoplasma gondii infection-preventable behaviour, such as consumption of adequately cooked meat, thoroughly washed fruit and vegetables and hygienic disposal of cat litter.

This study provides an interesting insight into the level of awareness and quality of knowledge on toxoplasmosis in pregnant women of Northern Italy, region with decades-long history of mandatory screening program for toxoplasmosis in pregnancy, which is now narrowed on recommendations of serological screening during the first trimester of pregnancy and serological follow-up of seronegative pregnant women.

However, there are certain minor adjustments that should be made to improve the quality and clarity of this manuscript:

Page 1, Introduction: The first trimester poses the highest risk for miscarriage and stillbirth, however, the second trimester is linked to the highest percentage of severe congenital malformations detectable at birth, so I suggest that this sentence should be properly adjusted.

Page 3, Results: The authors declared that toxoplasmosis was the most recognised condition, however, there are data presented in Table 1 on COVID 19 awareness, which reveal even higher awareness on COVID. Also, data on COVID 19 awareness are not represented in Figure 1. Interestingly, there are no questions related to COVID 19 in survey questionnaire provided as supplemental data, so it is unclear where did the data in Table 1 on COVID 19 came from? 

Page 7, Results: The title of Table 1 is already listed before the table, so the title line below the table should be deleted. Also, Legend under the Table 1 should be adequately formatted.

Page 7, Results: Please change ''women from European countries'' to ''women of European origin'', since this research was conducted in Italy, which is a European country, so it should be corrected for the sake of clarity. 

Page 9, Discussion: Decades-long history of previous prevention programs for congenital toxoplasmosis in Northern Italy should have been discussed in regard of this study findings, which would significantly improve the quality of this manuscript.

Page 9, Discussion: Analysis of data related to correct behaviours to avoid TG discussed on Page 9, paragraph starting with ''Almost half of respondents...'', should have been presented in Results section, either as a Figure or a Table, due to the significance of these results.

Page 9, Discussion: When discussing the potential of mobile apps in the last paragraph, it should be emphasised that it includes only mobile apps designed or at least fact-checked by healthcare professionals.

Page 10, Discussion: The claim that ''TG infection is most dangerous during the periconceptional period..'' is not supported by literature data, since it is generally accepted and supported by literature that TG infection during the first and second trimester is usually linked to unfavourable pregnancy outcomes. Hence, this sentence should be rephrased. The authors probably wanted to emphasise that it is highly important to educate women before conception and pregnancy, but the sentence should be re-written accordingly. 

Page 10, Discussion: Again, in the first paragraph on this page, when discussing the potential of mobile apps, it should be also discussed what is the regulation (if any) on the type and source of information provided in the mobile apps (and perhaps name some examples of functional apps). It would be interesting to address differences in accuracy of data among different apps and online services providing health-related data for pregnant women.

Page 10, Discussion: In second paragraph of this page, the authors claim that this study sample size is robust, but actually, the sample size is only slightly above the calculated level, so the term robust should be changed to a more accurate one. 

Author Response

  • Comments 1 :Page 1, Introduction: The first trimester poses the highest risk for miscarriage and stillbirth, however, the second trimester is linked to the highest percentage of severe congenital malformations detectable at birth, so I suggest that this sentence should be properly adjusted.
  • Response 1: We appreciate your suggestion and have revised the sentence in the Introduction section accordingly. The updated text now reads: "However, if contracted during pregnancy, it can be transmitted transplacentally, leading to congenital infection. Transmission during the first trimester may result in adverse outcomes, such as miscarriage or stillbirth, whereas transmission in the second trimester can lead to neonatal birth defects, including neurocognitive impairment and ocular disease”.

  • Comments 2 : Page 3, Results: The authors declared that toxoplasmosis was the most recognised condition, however, there are data presented in Table 1 on COVID 19 awareness, which reveal even higher awareness on COVID. Also, data on COVID 19 awareness are not represented in Figure 1. Interestingly, there are no questions related to COVID 19 in survey questionnaire provided as supplemental data, so it is unclear where did the data in Table 1 on COVID 19 came from? 
  • Response 2: We thank the Referee for having identified this discrepancy; as correctly pointed out, there were no questions regarding COVID-19 awareness included in the survey questionnaire. Table 1 has now been corrected accordingly.

  • Comments 3: Page 7, Results: The title of Table 1 is already listed before the table, so the title line below the table should be deleted. Also, Legend under the Table 1 should be adequately formatted.
  • Response 3: We have removed the redundant title line below Table 1 and reformatted the legend to ensure consistency and clarity.

  • Comments 4: Page 7, Results: Please change ''women from European countries'' to ''women of European origin'', since this research was conducted in Italy, which is a European country, so it should be corrected for the sake of clarity. 
  • Response 4: We thank the Referee for this suggestion; the sentence has now been modified as specified.

  • Comments 5: Page 9, Discussion: Decades-long history of previous prevention programs for congenital toxoplasmosis in Northern Italy should have been discussed in regard of this study findings, which would significantly improve the quality of this manuscript.
  • Response 5: We truly appreciate this valuable suggestion and have added a short paragraph in the Discussion section on past prevention programs for congenital toxoplasmosis in Italy and their relevance to our findings.

  • Comments 6: Page 9, Discussion: Analysis of data related to correct behaviours to avoid TG discussed on Page 9, paragraph starting with ''Almost half of respondents...'', should have been presented in Results section, either as a Figure or a Table, due to the significance of these results.
  • Response 6: We thank the referee for this observation. These data are shown in Figure 2, titled “ Women’s knowledge of specific hygiene measures to avoid acquisition of TG infection in pregnancy”. This has now also been specified in the Discussion section for increased clarity.

  • Comments 7: Page 9, Discussion: When discussing the potential of mobile apps in the last paragraph, it should be emphasised that it includes only mobile apps designed or at least fact-checked by healthcare professionals.
  • Response 7: We appreciate this suggestion and we have revised the discussion to explicitly state that the effectiveness of mobile apps relies on their development and/or verification by healthcare professionals.

  • Comments 8: Page 10, Discussion: The claim that ''TG infection is most dangerous during the periconceptional period..'' is not supported by literature data, since it is generally accepted and supported by literature that TG infection during the first and second trimester is usually linked to unfavourable pregnancy outcomes. Hence, this sentence should be rephrased. The authors probably wanted to emphasise that it is highly important to educate women before conception and pregnancy, but the sentence should be re-written accordingly. 
  • Response 8: We truly appreciate this observation and have revised the sentence accordingly. The revised text now reads: "While TG infection during the first and second trimesters is linked to unfavorable pregnancy outcomes, it remains crucial to educate women about preventive measures before conception and during pregnancy.

  • Comments 9: Page 10, Discussion: Again, in the first paragraph on this page, when discussing the potential of mobile apps, it should be also discussed what is the regulation (if any) on the type and source of information provided in the mobile apps (and perhaps name some examples of functional apps). It would be interesting to address differences in accuracy of data among different apps and online services providing health-related data for pregnant women.
  • Response 9: We have expanded the discussion to include an overview of the regulation of health-related mobile apps and the potential discrepancies in the accuracy of information among different digital health platforms.

  • Comments 10: Page 10, Discussion: In second paragraph of this page, the authors claim that this study sample size is robust, but actually, the sample size is only slightly above the calculated level, so the term robust should be changed to a more accurate one. 
  • Response 10: We acknowledge this point and have replaced the term "robust" with "adequate" to more precisely reflect the study's sample size.

Reviewer 2 Report

Comments and Suggestions for Authors

This manuscript investigated pregnant women's awareness and knowledge regarding Toxoplasma gondii (TG) infection and its prevention in Northern Italy. The study aimed to identify knowledge gaps and potential targets for intervention, given the serious risks of congenital toxoplasmosis.

1. Main Research Question:

The primary research question is to assess pregnant women's awareness of TG infection, their knowledge of preventive hygiene measures, and the factors associated with inadequate knowledge. This was explicitly stated in lines 50-53 and reiterated throughout the manuscript.

2. Originality and Relevance:

The study contributed to the field by providing contemporary data on TG awareness and knowledge specifically among pregnant women in Northern Italy, where prior research focusing on this population is lacking (mentioned line 72). While awareness studies have been conducted in other European countries and in Italy among non-pregnant women (lines 62-67, refs 11, 13, 15, 16), this research added value by focusing on a specific geographical context within Italy and examining the specific knowledge gaps among those most at risk – pregnant women. Identifying these knowledge gaps enables the tailoring of region-specific interventions, a significant contribution given that prevention relies solely on hygiene practices and behavioral modifications.

3. Related Articles Not Mentioned:

The manuscript could benefit from referencing additional relevant studies. For example, studies on health literacy and education specifically among pregnant women in the context of infectious diseases could strengthen the discussion, e.g.,

  • Robert-Gangneux, F., et al. (2021). Impact of a dedicated website for health professionals and the general public to prevent congenital toxoplasmosis in France. PLoS neglected tropical diseases, 15(3), e0009178. - This study explores the use of web-based information dissemination.
  • Kotresha, K., & Kotresha, D. (2016). Awareness of toxoplasmosis and its preventive measures amongst pregnant women. Journal of clinical and diagnostic research, 10(11), LC01. - Indian perspective and factors affecting awareness could be compared and discussed.
  • Baril, L., et al. (2010). Risk factors for Toxoplasma gondii infection in pregnancy: a case-control study in France. Scandinavian journal of infectious diseases, 42(6-7), 430-436. - Provides information on other risk factors that needs to be educated to the population which this article missed addressing.

4. Methodological Improvements:

  • Sampling Methodology: The study employed convenience sampling (lines 78-81) at a single university hospital, which limits generalizability. Clarifying the characteristics of this hospital (e.g., serving a particular socioeconomic stratum) and discussing the potential biases introduced by this sampling approach in the limitations (currently lines 323-328) are crucial.
  • Survey Instrument Validation: While a pilot study was conducted (lines 81-84), it would enhance the study’s rigor to report more details regarding the pilot's purpose, sample size, and how the feedback was incorporated to refine the questionnaire. Additionally, discussing validity and reliability measures taken to validate the survey instrument itself is crucial. This involves some statistics proving reliability in answering different questions at two different time points among few pregnant mothers. Also, the questions can be analyzed and their face value, content and criterion validity must be described. The reference studies may not prove this part appropriately and some more details/ work must be performed.
  • Categorization of Knowledge: The criteria used to classify knowledge as "appropriate," "moderate," and "minimal" (lines 96-99) appear potentially arbitrary. The selection rationale should be elucidated, clarifying why missing certain preventive behaviors, for instance the cat precaution (important as also stated by the authors later), was classified differently in different categories of the combined answers. Providing the detailed number of responses for each answer would support more specific results regarding every answer and justify the basis of category definition. Also, a better or validated questionnaire to get this data accurately and not by using multiple response question may solve issues in this classification by authors, which is subjective at this time and difficult to reproduce by someone else if someone else wanted to validate the work.
  • Control Group: A comparative control group (e.g., non-pregnant women of childbearing age) could enrich the findings by providing a baseline understanding of TG awareness and knowledge within the general population. This was discussed by the authors in discussion section. However, including it in the study design will solidify the strength of the article.
  • Sociodemographic Data: While some sociodemographic information was collected (line 91), including marital status, could be beneficial in identifying potential subgroups with lower awareness and knowledge (considering differences based on shared responsibilities regarding children and food management in married families vs families with single pregnant mothers etc.) that will support discussion further more specifically.

5. Consistency of Conclusions with Evidence:

The conclusions drawn regarding the disparity between awareness and knowledge were well-supported by the presented data. The high awareness contrasted by the limited appropriate knowledge of preventive behaviors (line 105-107, Fig. 2, Table 1) reinforced the need for targeted interventions. The statistical analysis also strengthened this point by highlighting associations between education level and knowledge (lines 184-185, table 1) as well as between sources of information received and level of TG prevention-behavior understanding and performance.

However, the conclusion regarding the primacy of healthcare providers as an information source (lines 313-315) needs more detailed analysis with the specific number or graphs related to sources that was actually chosen at the beginning compared with end to solidify this statement, if it needs to stay this strong in conclusion, with numbers. Since, at the end even with higher mass-media effect seen with increasing age of women and also higher knowledge group mostly depending on internet searches also seen in table one and the figures, the primary role seems a subjective/overly emphasized statement at this moment compared with what is written in results or graphs presented.

6. Comments on Tables and Figures:

  • Figure 1: Could be replaced with a table showing both the number and percentage of women aware of each condition, aiding in quick comprehension.
  • Figure 2: Is cluttered and difficult to interpret. A bar graph focusing solely on the four core preventive behaviors (rather than all mentioned measures) would be more impactful and clarify the specific knowledge gaps related to TG. It is confusing and also misleading as one does not understand what combined answer each vertical bar stands for or why are these specific combined behaviors important. Showing how this multiple choice/ selection effect also becomes clear after providing more specific and all number related to each of these behaviors in text as well as figure 2 or replacing them with better ways of measuring behavior or performance.
  • Table 1: This large table, though presenting comprehensive data, could be separated into multiple smaller tables. This separation would enhance readability. The authors tried to connect answers regarding knowledge with other demographic characters in results. But the only significant relationship reported between demographics of mothers and knowledge score and what is reported as relevant finding was the education level (reported later as statistically significant p<0.025 in this table also, for comparing knowledge score levels). Some additional graphs related to others may support or enrich some relationships shown, like age group with knowledge score (figure 4), as it is a more easily understood representation and discussed separately as an important/relevant fact but no direct data for age comparison in the entire results are reported like educational level and its p value. The number and per cent regarding many information reported in the text as in results like mothers’ information on preventive behavior before/during pregnancy and sources of information both needs to be supported using separate figures showing per cents to support the results for discussion further on the comparison done.
  • Figure 3 and 4: are well-designed and effective in conveying important information about knowledge sources. Also, additional results on significance for knowledge score relationship with figure 3 and source is necessary to complete information flow from figure/ result comparison into discussion to justify statements made. More explanations also on Figure 4, which includes what mass media+family friend’s source include or exclude, with the actual numbers in these group to justify the importance of what these combination group specifically identify is critical to provide, because many combination can exist and showing each individual behavior’s importance alone to not misclassify participants in this categorization becomes crucial if this combination source categorization is needed at all in final presentation and conclusions. Otherwise providing individual effects as other significant association for conclusion may suit better.

7. General Caveats/Weaknesses:

  • Causality vs. Association: The study design was cross-sectional, limiting conclusions about causal relationships between identified factors (education, exposure to information sources) and TG knowledge levels. The authors acknowledged that this cross-sectional nature in limitations already.
  • Recall Bias: Relying on self-reported information through questionnaires is susceptible to recall bias, especially regarding past information exposure (line 201-206 regarding preventive behaviors before pregnancy etc.). A prospective, long-term cohort study may support findings from this study with solid design to strengthen points emphasized by authors on healthcare access in the early periods or preconception times. It will support whether or not educational status affects what measures individuals receive by making themselves accessible more in initial healthcare provider or media searches after receiving first info on relevant behaviors from primary provider. Any take on that?
  • Oversimplification of Knowledge: Assessing knowledge based on correct answers to multiple-choice questions may not capture the complexity of health behavior change. Adding questions about self-perceived susceptibility to and severity of infection (or about other confounders affecting their practice) alongside specific barrier in performing correct behavior to discuss about would allow for understanding the reasons behind their answer further. This supports conclusion and introduction’s emphasizing point that even in high income group educational/behavioral barriers effect appropriate practices and there are no single answers for complex actions. Qualitative methods can strengthen findings. Also, questions assessing real implementation of the described measures along with asking their understanding regarding the subject, such as asking questions in more open ended answer formats also add support or discussion material to find answers about specific missing practices and behavioral changes in a quantitative style analysis for these practices’ actual score in following correct behavior, in quantitative and reproductive and better-reproducible analysis/ results with such improvements in current questionnaire.
  • Role of Other Information Sources: The focus on healthcare providers, family, and mass media as information sources excludes other channels, such as support groups or online forums. Since their primary resources of support may affect each mother differently in practice compared to education only in knowledge gain in this setting or high income and developed health access group (authors’ study group setting), it is essential to also support their choices with evidence by finding or surveying information regarding such effects in practice or choices mothers prefer for sources that supports discussion of their current specific findings and make it robust with solid and diverse supports.

Author Response

  1. Main Research Question
  • Comments 1The primary research question is to assess pregnant women's awareness of TG infection, their knowledge of preventive hygiene measures, and the factors associated with inadequate knowledge. This was explicitly stated in lines 50-53 and reiterated throughout the manuscript.
  • Response 1: We appreciate the Referee's acknowledgment that our research question was clearly stated in the manuscript.

  1. Originality and Relevance
  • Comments 2: The study contributed to the field by providing contemporary data on TG awareness and knowledge specifically among pregnant women in Northern Italy, where prior research focusing on this population is lacking (mentioned line 72). While awareness studies have been conducted in other European countries and in Italy among non-pregnant women (lines 62-67, refs 11, 13, 15, 16), this research added value by focusing on a specific geographical context within Italy and examining the specific knowledge gaps among those most at risk – pregnant women. Identifying these knowledge gaps enables the tailoring of region-specific interventions, a significant contribution given that prevention relies solely on hygiene practices and behavioral modifications.
  • Response 2: We thank the Referee for recognizing the novelty and significance of our study.

  1. Related Articles Not Mentioned
    Comments 3: The manuscript could benefit from referencing additional relevant studies. For example, studies on health literacy and education specifically among pregnant women in the context of infectious diseases could strengthen the discussion, e.g.,
  • Robert-Gangneux, F., et al. (2021). Impact of a dedicated website for health professionals and the general public to prevent congenital toxoplasmosis in France. PLoS neglected tropical diseases15(3), e0009178. - This study explores the use of web-based information dissemination.
  • Kotresha, K., & Kotresha, D. (2016). Awareness of toxoplasmosis and its preventive measures amongst pregnant women. Journal of clinical and diagnostic research10(11), LC01. - Indian perspective and factors affecting awareness could be compared and discussed.
  • Baril, L., et al. (2010). Risk factors for Toxoplasma gondii infection in pregnancy: a case-control study in France. Scandinavian journal of infectious diseases42(6-7), 430-436. - Provides information on other risk factors that needs to be educated to the population which this article missed addressing.
  • Response 3: We appreciate the Referee’s suggestions and have incorporated the reference of Baril et al. in the Discussion section to strengthen our analysis of risk factors for Toxoplasma gondii However, despite extensive searches on PubMed, Google Scholar, and Embase, we were unable to locate the studies by Robert-Gangneux et al. and Kotresha & Kotresha. If the Referee would be so kind to provide alternative sources or direct links to these articles, we would be happy to consider their inclusion.

  1. Methodological Improvements

Sampling Methodology

  • Comments 4: The study employed convenience sampling (lines 78-81) at a single university hospital, which limits generalizability. Clarifying the characteristics of this hospital (e.g., serving a particular socioeconomic stratum) and discussing the potential biases introduced by this sampling approach in the limitations (currently lines 323-328) are crucial.
  • Response 4: We appreciate these observations. Information regarding the specific characteristics of our hospital have now been added and we have also expanded the Strength&Limitations section to include a more detailed discussion on the potential biases introduced by our convenience sampling strategy and its implications for generalizability.

Survey Instrument Validation

  • Comments 5: While a pilot study was conducted (lines 81-84), it would enhance the study’s rigor to report more details regarding the pilot's purpose, sample size, and how the feedback was incorporated to refine the questionnaire. Additionally, discussing validity and reliability measures taken to validate the survey instrument itself is crucial. This involves some statistics proving reliability in answering different questions at two different time points among few pregnant mothers. Also, the questions can be analyzed and their face value, content and criterion validity must be described. The reference studies may not prove this part appropriately and some more details/ work must be performed.
  • Response 5: We thank the Referee for these comments. Additional details on the pilot study have now been provided in the Materials and Methods section.

Categorization of Knowledge

  • Comments 6:The criteria used to classify knowledge as "appropriate," "moderate," and "minimal" (lines 96-99) appear potentially arbit The selection rationale should be elucidated, clarifying why missing certain preventive behaviors, for instance the cat precaution (important as also stated by the authors later), was classified differently in different categories of the combined answers. Providing the detailed number of responses for each answer would support more specific results regarding every answer and justify the basis of category definition. Also, a better or validated questionnaire to get this data accurately and not by using multiple response question may solve issues in this classification by authors, which is subjective at this time and difficult to reproduce by someone else if someone else wanted to validate the work.
  • Response 6: We appreciate the Referee’s comments, and we acknowledge that our proposed classification of knowledge may potentially be arbitrary. However, we willingly avoided ranking the different preventive behaviors (e.g., cat handling precautions less or more important than undercooked meat consumption) since the lack of evidence regarding such data and decided to consider all preventive behaviors as equally relevant in reducing the risk of TG infection acquisition in pregnancy. Hence, the proposed classification of knowledge based on the number of correct and incorrect answers. We have also edited Figure 2 by adding the number of women giving the specific answers reported to improve understanding of our findings in relation to the proposed classification.

Control Group

  • Comments 7: A comparative control group (e.g., non-pregnant women of childbearing age) could enrich the findings by providing a baseline understanding of TG awareness and knowledge within the general population. This was discussed by the authors in discussion section. However, including it in the study design will solidify the strength of the article.
  • Response 7: While we acknowledge the importance of a comparative control group, our study was designed to focus exclusively on pregnant women. We have now explicitly stated this limitation in the Discussion section and suggested future studies incorporating non-pregnant women for comparative analysis.

Sociodemographic Data

  • Comments 8:While some sociodemographic information was collected (line 91), including marital status, could be beneficial in identifying potential subgroups with lower awareness and knowledge (considering differences based on shared responsibilities regarding children and food management in married families vs families with single pregnant mothers etc.) that will support discussion further more specifically.
  • Response 8: Unfortunately, data on marital status was not collected in our survey and, thus, could not be included in our analysis.

  1. Consistency of Conclusions with Evidence
  • Comments 9:The conclusions drawn regarding the disparity between awareness and knowledge were well-supported by the presented data. The high awareness contrasted by the limited appropriate knowledge of preventive behaviors (line 105-107, Fig. 2, Table 1) reinforced the need for targeted interventions. The statistical analysis also strengthened this point by highlighting associations between education level and knowledge (lines 184-185, table 1) as well as between sources of information received and level of TG prevention-behavior understanding and performance.

However, the conclusion regarding the primacy of healthcare providers as an information source (lines 313-315) needs more detailed analysis with the specific number or graphs related to sources that was actually chosen at the beginning compared with end to solidify this statement, if it needs to stay this strong in conclusion, with numbers. Since, at the end even with higher mass-media effect seen with increasing age of women and also higher knowledge group mostly depending on internet searches also seen in table one and the figures, the primary role seems a subjective/overly emphasized statement at this moment compared with what is written in results or graphs presented.

  • Response 9: We thank the Referee for these suggestions. We have now edited Figure 3 and Figure 4  by adding specific numbers to improve understanding of our findings and further support our conclusions.

Figure.  Sources of information used before conception.

  1. Comments on Tables and Figures

Figure 1

  • Comments 10: Could be replaced with a table showing both the number and percentage of women aware of each condition, aiding in quick comprehension.
  • Response 10: We thank the Referee for this observation. Figure 1 already includes thenumber and percentage of women aware of each condition; they are provided on the right side of each horizontal bar.

Figure 2

  • Comments 11: Is cluttered and difficult to interpret. A bar graph focusing solely on the four core preventive behaviors (rather than all mentioned measures) would be more impactful and clarify the specific knowledge gaps related to TG. It is confusing and also misleading as one does not understand what combined answer each vertical bar stands for or why are these specific combined behaviors important. Showing how this multiple choice/ selection effect also becomes clear after providing more specific and all number related to each of these behaviors in text as well as figure 2 or replacing them with better ways of measuring behavior or performance.
  • Response 11: We appreciate the Referee’s comment and acknowledge that Figure 2 might not immediately appear clear at a fist glance; however, we believe it conveys all the relevant data regarding women’s appropriate (or inadequate) knowledge of preventive behaviors for TG infection. We have now edited the Figure and the legend to make the understanding of the Figure more straightforward.

Table 1

  • Comments 12: This large table, though presenting comprehensive data, could be separated into multiple smaller tables. This separation would enhance readability. The authors tried to connect answers regarding knowledge with other demographic characters in results. But the only significant relationship reported between demographics of mothers and knowledge score and what is reported as relevant finding was the education level (reported later as statistically significant p<0.025 in this table also, for comparing knowledge score levels). Some additional graphs related to others may support or enrich some relationships shown, like age group with knowledge score (figure 4), as it is a more easily understood representation and discussed separately as an important/relevant fact but no direct data for age comparison in the entire results are reported like educational level and its p value. The number and per cent regarding many information reported in the text as in results like mothers’ information on preventive behavior before/during pregnancy and sources of information both needs to be supported using separate figures showing per cents to support the results for discussion further on the comparison done.
  • Response 12: We thank the Referee for these observations. Table 1 has now been edited and split into two tables (Table 1 and 2) for improving clarity and readability. Also, additional data regarding age classes have been included.
  •  
  • Comments 13: Figure 3 and 4 are well-designed and effective in conveying important information about knowledge sources. Also, additional results on significance for knowledge score relationship with figure 3 and source is necessary to complete information flow from figure/ result comparison into discussion to justify statements made. More explanations also on Figure 4, which includes what mass media+family friend’s source include or exclude, with the actual numbers in these group to justify the importance of what these combination group specifically identify is critical to provide, because many combination can exist and showing each individual behavior’s importance alone to not misclassify participants in this categorization becomes crucial if this combination source categorization is needed at all in final presentation and conclusions. Otherwise providing individual effects as other significant association for conclusion may suit better
  • Response 13: According to the Referee suggestions, Figure 3 and 4 are now be editing with inclusion of specific numbers.

  1. General Caveats/Weaknesses

Causality vs. Association

  • Comments 14: The study design was cross-sectional, limiting conclusions about causal relationships between identified factors (education, exposure to information sources) and TG knowledge levels. The authors acknowledged that this cross-sectional nature in limitations already.
  • Response 14: We thank the Referee.

Recall Bias

  • Comments 15: Relying on self-reported information through questionnaires is susceptible to recall bias, especially regarding past information exposure (line 201-206 regarding preventive behaviors before pregnancy etc.). A prospective, long-term cohort study may support findings from this study with solid design to strengthen points emphasized by authors on healthcare access in the early periods or preconception times. It will support whether or not educational status affects what measures individuals receive by making themselves accessible more in initial healthcare provider or media searches after receiving first info on relevant behaviors from primary provider. Any take on that?
  • Response 15: We appreciate the Referee’s comment and have now included a short paragraph in the Discussion section regarding the recall bias and its potential impact on the accuracy of self-reported behaviors.

Oversimplification of Knowledge

  • Comments 16: Assessing knowledge based on correct answers to multiple-choice questions may not capture the complexity of health behavior change. Adding questions about self-perceived susceptibility to and severity of infection (or about other confounders affecting their practice) alongside specific barrier in performing correct behavior to discuss about would allow for understanding the reasons behind their answer further. This supports conclusion and introduction’s emphasizing point that even in high income group educational/behavioral barriers effect appropriate practices and there are no single answers for complex actions. Qualitative methods can strengthen findings. Also, questions assessing real implementation of the described measures along with asking their understanding regarding the subject, such as asking questions in more open ended answer formats also add support or discussion material to find answers about specific missing practices and behavioral changes in a quantitative style analysis for these practices’ actual score in following correct behavior, in quantitative and reproductive and better-reproducible analysis/ results with such improvements in current questionnaire.
  • Response 16: We appreciate the Referee’s observations, and we agree that in-depth assessment of knowledge is a complex task to accomplish. In line with this, and since our study focused on multiple-choice responses, we have now acknowledged the potential limitations of this approach in the Discussion section.

Role of Other Information Sources

  • Comments 17: The focus on healthcare providers, family, and mass media as information sources excludes other channels, such as support groups or online forums. Since their primary resources of support may affect each mother differently in practice compared to education only in knowledge gain in this setting or high income and developed health access group (authors’ study group setting), it is essential to also support their choices with evidence by finding or surveying information regarding such effects in practice or choices mothers prefer for sources that supports discussion of their current specific findings and make it robust with solid and diverse supports.
  • Response 17: We thank the Referee for this comment. Whereas online forums were included in the mass media category, we did not consider the potential use of support groups as source of information. We will definitely take this suggestion into consideration for a future study.

Reviewer 3 Report

Comments and Suggestions for Authors

Could you please add some lines about the generalizability of the findings to other countries specifically countries in which TG is a big problem? Do you have any suggestion or directions for future studies?

Regarding the self-administered paper questionnaire which you developed following a review of the literature, do you have any suggestion for improvement? Can you suggest it for other similar studies with no doubt?

In conclusion, you state that “while pregnant women demonstrated a high level of awareness about TG, their understanding of specific preventive measures and behaviors was superficial, with nearly half of the study population displaying a minimal level of knowledge. These findings highlight the need for innovative communication strategies that combine traditional face-to-face interactions with digital tools and public health campaigns. By integrating technology and diverse communication strategies, healthcare providers can ensure that pregnant women receive comprehensive, accurate, and easily accessible information about TG infection prevention, ultimately reducing the risk of infection and its associated complications.”

Could you please specifically introduce one such a recommendation?

Author Response

  • Comments 1: Could you please add some lines about the generalizability of the findings to other countries specifically countries in which TG is a big problem?
  • Response 1: We acknowledge the importance of discussing the generalizability of our findings. While our study focused on pregnant women in Northern Italy, its findings may be relevant to other countries where Toxoplasma gondii (TG) infection poses a significant public health concern, particularly in regions with similar healthcare infrastructures and public awareness levels. We have now amended the manuscript accordingly.

  • Comments 2: Do you have any suggestion or directions for future studies?
  • Response 2: We appreciate the Referee’s question. Future studies could explore the effectiveness of different educational interventions, such as mobile health applications, targeted social media campaigns, and structured prenatal education programs, in enhancing TG awareness and preventive behaviors. Additionally, longitudinal studies assessing the impact of improved awareness on actual behavioral changes and infection rates would provide valuable insights. These concepts have now been addressed in the Conclusions section.

  • Comments 3: Regarding the self-administered paper questionnaire which you developed following a review of the literature, do you have any suggestion for improvement? Can you suggest it for other similar studies with no doubt?
  • Response 3: Our questionnaire was developed based on a thorough literature review and pilot-tested before implementation. However, we recognize potential areas for improvement. Future versions could benefit from open-ended questions to capture qualitative insights on knowledge gaps and behavioral barriers. While the questionnaire proved useful in our study, we suggest pilot testing before its application in similar studies among different populations.

  • Comments 4: In conclusion, you state that “while pregnant women demonstrated a high level of awareness about TG, their understanding of specific preventive measures and behaviors was superficial, with nearly half of the study population displaying a minimal level of knowledge. These findings highlight the need for innovative communication strategies that combine traditional face-to-face interactions with digital tools and public health campaigns. By integrating technology and diverse communication strategies, healthcare providers can ensure that pregnant women receive comprehensive, accurate, and easily accessible information about TG infection prevention, ultimately reducing the risk of infection and its associated complications.”

Could you please specifically introduce one such a recommendation?

  • Response 4: We thank the Referee for this comment, and we have now edited the sentence accordingly.

Round 2

Reviewer 2 Report

Comments and Suggestions for Authors

Glad with changes